# From static to dynamic: Embracing dynamics in isotopic diet estimation

Emilie Cathelin[1]☺*, Sebastien Lefebvre[1]☺, Carolina Giraldo[2]☺

**1** Station marine de Wimereux, UMR 8187 – LOG – Laboratoire d'Océanologie et de Géosciences, Université de Lille, CNRS, Université Littoral Côte d'Opale, IRD, Lille, France, **2** HMMN – Unité halieutique Manche-Mer du Nord, IFREMER, Boulogne sur mer, France

☺ These authors contributed equally to this work.
* emilie00cathelin@gmail.com

## Abstract

Isotopic mixing models are widely used in ecology to quantify the diets of organisms. Most of these models assume that stable isotopic systems remain stable over time (a premise known as the *steady-state hypothesis*) and are therefore referred to as static mixing models. However, evidence shows that temporal dynamics—such as variations in isotopic turnover rates, diet shifts, and fluctuations in the isotopic signatures of both sources and consumers—can introduce significant bias into model outputs. Despite the recognition of such dynamics, the factors influencing bias and its implications remain underexplored. This study uses modelling and *in silico* experiments to characterize bias in mixing models resulting from temporal dynamics and to develop a dynamic mixing model that accounts for these effects. The results revealed that bias is strongly influenced by the interaction between the isotopic turnover rate and sampling frequency, emphasizing the importance of time variability in isotopic turnover. Additional sources of bias include the consumer's isotopic signature prior to a dietary shift, which reflects the distance from equilibrium with the new diet. This bias can be further amplified by temporal fluctuations in source signatures. Our study also evaluates previously recommended strategies to mitigate bias, confirming their effectiveness in reducing errors in static models and providing additional guidelines for their application. Furthermore, it improves access to a dynamic mixing model, enabling direct comparison with static approaches and demonstrating its robustness and accuracy in estimating diets under dynamic conditions.

## Introduction

The study of isotopic composition in tissues is a widely used technique in ecology [1]. This study provides insights into trophic levels and food web structure [2–6], as well as the origin of carbon in trophic webs [4]. Stable isotope analysis can also be used to study the diets of organisms or identify pollution sources

**Data availability statement:** All codes and data files are available in Zenodo: https://doi.org/10.5281/zenodo.16040315 the provided scripts support deterministic replication of the analyses but do not implement uncertainty propagation or MCMC sampling.

**Funding:** This work was financed by the PhD fellowship (EC) granted by the Region Hauts-de-France and the Graduate School EDSMRE from University of Lille. The funders had no role in study design, data collection and analysis, decision to publish, or preparation of the manuscript.

**Competing interests:** The authors have declared that no competing interests exist.

through mixing models (MMs) [7,8] (refer to Table 1 for acronyms and notations). These models are based on the principle of "You are what you eat (plus a few ‰)" [9]. When consumer diets are studied, the "plus a few ‰" difference between the consumer's isotopic signature and the signatures of its food sources arises from the differential use of light and heavy isotopes by the consumer's metabolic pathways. This difference is reflected in two key parameters. First, the trophic discrimination factor (TDF, $\Delta X_s$) represents the fractionation between the isotopic signature of the ingested diet and its corresponding signature in the consumer's tissues once equilibrium is reached with respect to that diet [4,10–12]. The TDF is incorporated into MMs to correct for isotopic source signatures. Second, there is a delay (or incorporation time window) between the onset of a new diet and the consumer's full tissue equilibrium, which depends on the isotopic turnover rate (λ). λ varies depending on the tissue type [10] and can also be expressed as a function of the half-life ($t_{\frac{1}{2}}$), which is the time required for a tissue to change half of its isotopic composition ($\lambda = \frac{\log(2)}{t_{\frac{1}{2}}}$). λ has only recently been incorporated into MMs using a temporal approach, requiring the sampling of both sources and consumers over a defined time window, which is a function of λ [13].

**Table 1. Acronyms and notations used in the study.**

| Acronyms | Definition |
|---|---|
| MM | Mixing Model |
| SMM | Static Mixing Model |
| SMMΔ | Integrated Mixing Model (*Static Mixing Model with Temporal Integration*) |
| DMM | Dynamic Mixing Model |
| **Notations** | **Definition** |
| S | Number of food sources |
| I | Number of isotopes |
| TDF, $\Delta X_i$ | Trophic discrimination factor of source $i$ (‰) |
| λ | Isotopic turnover rate (d$^{-1}$) |
| $t$ | Time (d) |
| T | Time window between two samplings (d) |
| $\tau$ | Time window (twice the half-life) (d) |
| $t_{\frac{1}{2}}$ | Isotopic half-life (d) |
| $q_i$ | Elemental concentration in source i |
| $p_i$ | Dietary contribution of source $i$ |
| $\hat{p}_i$ | Estimated dietary contribution of source $i$ |
| $\delta X_i$ or $\delta Y_i$ | Isotopic signature of source $i$ in isotope $X$ or $Y$ (‰) |
| $\delta X$ or $\delta Y$ | Isotopic signature of the consumer in isotope $X$ or $Y$ (‰) |
| $\langle \delta X_i(\tau) \rangle$ | Time-averaged isotopic signature of source $i$ over window $\tau$ |
| $\hat{\delta X}$ | Estimated isotopic signature of the consumer in isotope $X$ (‰) |
| $\delta X(t), \delta X(\infty), \delta X(0)$ | Isotopic signature of the consumer in isotope X (‰) at time $t$, when $t$ tends to $\infty$, and at $t = 0$, respectively |
| $\delta X_{eq}$ | Consumer signature in isotope X when at equilibrium (‰) |
| $\beta$ | Relative system bias (%) |

Although both TDF and λ were described over 40 years ago and are recognized as important in the dynamics of isotope incorporation [10], they are treated differently in the literature because of challenges in estimation and sampling. TDF is widely acknowledged as critical for accurate diet estimation and is well studied [14–16]; however, λ has received much less attention in diet studies. This parameter is absent from commonly used mixing models, such as SIAR, SIMMr and MixSIAR [17–19]. Both source and consumer signatures are subject to temporal dynamics, as observed in field data [20] and demonstrated in theoretical or *in silico* models [13,21–23]. These dynamics can lead to errors in diet estimation when static mixing models (SMMs) are used. Ballutaud et al. [13] reported that diet estimation errors in a two-source system can reach 50%. However, these errors have been minimally investigated, and the specific factors contributing to these biases have been only partially described [22] or quantified [13]. To reduce these biases, recommendations and best practices for using static models have been proposed through an integrated SMM (SMMΔ) [22]. This model accounts for temporal variability in source signatures by averaging them over a time window before measuring the consumer signatures, thus reducing estimation bias. To illustrate the disparity in the treatment of TDF and λ, we reviewed 103 scientific articles that used mixing models to estimate diets, published between 2014 (the year Phillips et al. [22] introduced SMMΔ) and 2024. Using the keywords "isotopic," "mixing models," "MixSIAR," "ecology," and "diet" (details in S1 Appendix), our analysis revealed that 87% of the articles actively estimated TDF, whereas only 26% mentioned λ, and only 4% actually used λ to sample source signatures accordingly. This suggests that λ is significantly underrepresented in the field.

The issue with omitting λ in mixing models is that it implicitly assumes that the system under study is at equilibrium (the steady-state hypothesis). In other words, it assumes that the system is in a steady state, with no temporal changes in any component, such as the isotopic signatures of sources and consumers, diet composition, λ, or TDF. In our review, 18% of the papers estimated at least some temporal dynamics or verified the system's stability, whereas 7% explicitly stated that they were working under the assumption of stability. This results in 75% implicitly assuming equilibrium without acknowledging or verifying it. Although this assumption may be valid in some cases, temporal changes in isotopic components—such as seasonal baseline shifts—have been observed [24–30]. These shifts affect the entire food web, albeit to a lesser extent as one moves up trophic levels [31], making the food web isotopically season dependent. Additionally, consumer isotopic signatures may fluctuate throughout the year [32–34], as a consumer's diet often changes throughout life due to factors such as feeding behaviour, ontogenetic changes [35], food availability, and migration patterns [36]. As a result, applying the steady-state hypothesis can sometimes be challenging or inappropriate.

There are two main approaches to modelling in mixing models. Historical models typically use a frequentist approach [7,13,37–40], which provides single-value estimates along with confidence intervals. Recent developments have shifted towards Bayesian approaches [17–19], such as MixSIAR, which offer probability distributions of estimates rather than point values. This Bayesian framework represents a significant advancement, particularly in its ability to manage uncertainties and errors in data, parameter estimations, and statistics [41,42]. However, Bayesian models do not facilitate model evaluation [43]. Furthermore, despite these improvements, existing Bayesian mixing models still do not incorporate λ, the isotopic turnover rate, which is crucial for accurately accounting for temporal dynamics in diet studies. Although these models handle source variability to some extent and work similarly to integrated static models (SMMΔ), they remain limited in their ability to fully capture the dynamic nature of isotopic incorporation. Notably, no current Bayesian MM can integrate the full range of temporal isotopic dynamics—particularly the role of λ—as proposed in this study.

To address these isotopic dynamics, the dynamic mixing model (DMM) was developed [13]. The DMM accounts for mixing over a time period rather than at a single point, allowing it to incorporate variability in both sources and consumers during the time window. However, its current application is limited to datasets with only one isotope and two food sources. In contrast, most diet studies involve two isotopes (typically carbon and nitrogen) and multiple food sources [27,44–46]. Additionally, the DMM does not account for concentration dependence across different components [38]. In Ballutaud et al. [13], an attempt was made to estimate the bias of static mixing models (SMM, SMMΔ), but the factor used to explain

the bias—empirically set as the frequency of diet switches divided by λ—could not be generalized or applied to field studies. This limitation makes it impossible to estimate the bias of an isotopic system with prior knowledge.

This study aims to generalize the DMM for broader use in isotopic diet studies, enabling its application with multiple isotopes and food sources. Furthermore, we seek to describe and quantify the biases associated with using static mixing models, particularly concerning the overlooked parameter λ. Last, a concrete case study will illustrate the practical application of the generalized DMM and demonstrate the impact of these biases, providing clearer guidelines for future isotopic diet studies.

## Materials and methods

### Mixing models

Because no existing Bayesian MM supports a dynamic mixing model (DMM), all modelling in this study was conducted using a frequentist approach, which is easier to implement and more widely accessible. Although the frequentist approach does not offer the flexibility and uncertainty quantification of Bayesian methods, it still provides a robust framework for comparing static and dynamic mixing models [13]. A Bayesian approach would have allowed for more sophisticated handling of model uncertainty and error structure [42]. However, despite these advantages, the frequentist method was sufficient for the purposes of comparing the two modelling approaches and testing the hypothesis of dynamic versus static isotopic incorporation.

The static mixing model (SMM) is based on the principle "You are what you eat (plus a few ‰)" [4,9]. It hypothesizes that the consumer's isotopic signature (expressed in ‰ for an element X, or $\delta X$) is derived from the isotopic signatures of its food sources ($\delta X_S$), weighted by their respective contributions to the diet ($p_S$), plus a fraction called the TDF ($\Delta X_s$). Because the model assumes that all potential food sources are accounted for, the sum of all dietary contributions equals one, ensuring that $\sum_{i=1}^{S} p_i = 1$. Further refinements to the model involve weighting the source signatures by their concentration of the studied element ($q_s$) for increased accuracy [17,38]. Thus, the consumer signature at equilibrium, $\delta X_{eq}$, can be expressed as shown in Equation (1). This static mixing model represents a snapshot of the system at a given time ($t$), assuming that the system is at equilibrium. Therefore, $\delta X_{eq} = \delta X(\infty) = \lim_{t \to +\infty} \delta X(t)$ regardless of $t$. This assumption implies that the food sources, diet composition, isotopic concentrations, and the consumer's isotopic signature remain constant over time.

$$\delta X_{eq} = \frac{\sum_{i=1}^{S} p_i q_i \left( \delta X_i + \Delta X_i \right)}{\sum_{i=1}^{S} p_i q_i}$$

(1)

To address the limitations identified by Phillips et al. [22], we implemented an integrated static model (SMMΔ), which builds on the SMM but differs in that the source signature values are averaged over a time window equal to twice the half-life of the element in the tissue ($2t_{\frac{1}{2}} = 2\frac{log(2)}{\lambda}$), as described in Ballutaud et al. [13]. The isotopic signature of the consumer can then be expressed as shown in Equation (2). This model is no longer a snapshot, following the guidelines proposed by Phillips et al. [22], and it incorporates λ as a parameter to calculate the time window. Here, $\langle \delta X_i(\tau) \rangle$ represents the average source signature over each time window $\tau$ within $[\, t - 2\frac{log(2)}{\lambda} \,, t\,]$. As in the SMM, the equilibrium is still defined as $\delta X_{eq} = \delta X(\infty)$.

$$\delta X_{eq} = \frac{\sum_{i=1}^{S} p_i q_i \left( \langle \delta X_i(\tau) \rangle + \Delta X_i \right)}{\sum_{i=1}^{S} p_i q_i} \; with \; \tau \in \langle \delta X_i(\tau) | \tau \in [\, t - 2\frac{log(2)}{\lambda} \,, t\,] \, \rangle$$

(2)

The third model takes a different approach. In studies examining diet switches, the change in a consumer's isotopic signature over time ($\delta X(t)$) is typically described by a first-order kinetic one-compartment time model, as shown in Equation (3) [40,47].

$$\delta X(t) = \delta X_{eq} + \left( \delta X(0) - \delta X_{eq} \right) e^{-\lambda t}$$

(3)

Where $\delta X(0)$ represents the consumer's isotopic signature at the beginning of the experiment or sampling period and where λ represents the isotopic turnover rate (in d$^{-1}$). This model is applicable when $\delta X_{eq}$ and λ are constant over time. However, in the DMM, the assumption is that $\delta X_{eq}$ follows Equation (1), but with $p_i$, $q_i$, $\delta X_i$ and $\Delta X_i$ vary over time (see Eq. 4). In this case, $\delta X_{eq}$ is no longer equal to $\delta X(\infty)$, as in the original model; instead, it becomes a time-dependent variable.

$$\delta X_{eq}(t) = \frac{\sum_{i=1}^{S} p_i(t) q_i(t) \left(\delta X_i(t) + \Delta X_i(t)\right)}{\sum_{i=1}^{S} p_i(t) q_i(t)}$$

(4)

The DMM uses an ordinary differential equation version of the time model (Eq. 5) [13,34].

$$\frac{d\delta X}{dt} = \lambda(t) \left(\delta X_{eq}(t) - \delta X\right)$$

(5)

Equation (3) provides an analytical solution to Equation (5) when no components vary over time. However, when source signatures or λ change over time, no analytical solution of Equation (3) exists, and Equation (5) is used to model how the consumer's isotopic signature changes ($\frac{d\delta X}{dt}$, Eq. 5). To estimate the consumer signature at each time point ($\delta X(t)$) during the period of interest, a numerical solver from the **deSolve** (ver 1.40, [48]) package is used. This method allows for easy incorporation of changes in both source signatures and λ over time.

The DMM is used to estimate the average diet of a consumer between two sampling dates, i.e., over a time window (T). The first sampling date is set as t = 0, marking the start of T (and any possible change in diet from a previous one), whereas the second sampling date represents the end of T. For longer time series with more than two sampling dates, the DMM is applied between each pair of sampling dates (i.e., each time window) to estimate the mean diet at each T.

The three models were developed through a similar process (see Fig 1) and are compatible with various combinations of studied isotopes and numbers of sources. Each model follows a procedure based on the IsoSource method [37], described as follows: First, all potential combinations of source contributions are generated, with the precision of the proportions (the increment) being controllable. For example, in a three-source system with a precision increment of 0.01‰, a total of 5,151 possible combinations of source contributions are generated. Next, one of the mixing models (SMM, SMMΔ, or DMM) is applied to predict the consumer's isotopic signature for each combination of source contributions. Equation (1) is used for the SMM, Equation (2) is used for the SMMΔ, and Equation (5) is used for the DMM. Afterwards, the user can choose to either select a subset of combinations that best fit the observed consumer signatures or retain all combinations that predict a consumer signature within a certain tolerance of the observed value. The quality of the fit between the observed ($\delta X$) and predicted ($\delta \hat{X}$) consumer signatures is assessed using the estimation error [49] (Eq. 6), where I represents the number of studied isotopes. The closer the estimation error is to 0, the better the model fits the data.

$$\text{estimation error} = \sqrt{\sum_{j=1}^{I} \left(\frac{\delta X_j - \delta \hat{X}_j}{\delta X_j}\right)^2}$$

(6)

All three models produce similar outputs, which are the best-fitting combinations of contributions (or proportions) for each source. However, the outputs represent the mean diet over different temporal scales for each model (see Fig 1). The SMM provides a "snapshot" estimation of the diet at the sampling time, assuming that equilibrium has been reached. The SMMΔ estimates the mean diet over a period equal to $2t_{\frac{1}{2}}$ days without assuming equilibrium. Finally, the DMM estimates the mean diet over an interval between two sampling times and can also capture the variation in the consumer's isotopic signature over this period. In summary, the SMM is suitable for estimating source contributions in systems where equilibrium is assumed, whereas the SMMΔ is more appropriate for systems where the sources vary but the consumer is stable.

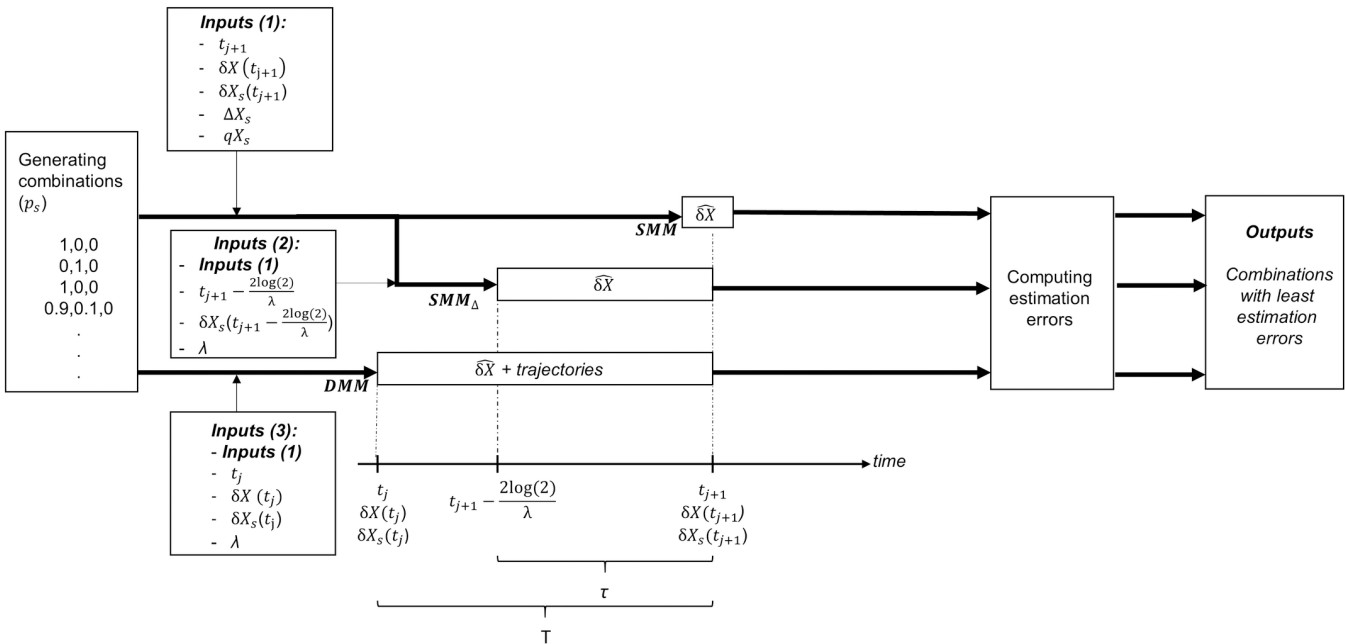

**Fig 1. Framework and operation for the SMM, SMMΔ and DMM (see Table 1 for definitions and acronyms).** The processes for the three models are depicted with dark lines, showing the required inputs, the diet estimation time (or period, depending on the model), and the process or the solution selection. i and i+1 represent two sampling dates.

Both the SMM and SMMΔ can be considered specific cases of the DMM, which can be applied regardless of whether the system is at equilibrium. The code was implemented in R (version 4.4.0), and all the scripts used to produce the results in this study are available on Zenodo (available at https://doi.org/10.5281/zenodo.16040315). The code provides functions to running all three mixing models using a deterministic approach, as well as several graphical outputs to ease interpretation of the results and to assess fittings. However, the model is implemented assuming no data or parameter-based uncertainty and does not address error structure arising from data uncertainty, nor does it include a formal diagnosis of contribution correlations which must be conducted separately.

### Bias in mixing models

We introduce the concept of bias in a mixing model, specifically focusing on the bias generated by the model itself in relation to temporal dynamics. For this study, all the data and parameters were considered "certain," meaning that no uncertainty was introduced into the analysis. The bias for a single source (denoted as $i$ in a system with S sources) is defined as the difference between its actual contribution to the diet (denoted $p_i$) and its estimated or predicted contribution by the model (denoted $\hat{p}_i$). To estimate the overall system bias, denoted as Bias, we sum the absolute value of the bias for each source (Eq. 7). This Bias ranges between 0 and 2, where 0 indicates no bias (i.e., the model accurately estimates the real diet) and 2 indicates complete bias (i.e., the model fails entirely to estimate the real diet).

$$\text{Bias} = \sum_{i=1}^{S} \left| p_i - \hat{p}_i \right| \tag{7}$$

For the remainder of this study, the bias is expressed as the relative bias of the system $\beta$ expressed in % of the maximal Bias (see Eq. 8).

$$\beta = \frac{Bias}{2} 100$$

<div style="text-align: right">(8)</div>

When the time model (Eq. 3) is used to model isotopic integration, three key parameters influence the bias in SMM estimates. First, the λt value plays a role in the bias, as higher values of λt result in lower bias. Second, the initial isotopic signature of the consumer at the beginning of the time period ($\delta X(0), \delta Y(0)$) affects the bias, a phenomenon referred to as the "switch effect." This effect can also be interpreted as the change in the consumer's isotopic signature from the start of the experiment to equilibrium. The switch effect is quantified by estimating the Euclidean distance between the initial signature and equilibrium relative to the size of the isotopic space. The Euclidean distance between the initial signature and equilibrium, divided by the longest distance in the source polygon, is referred to as the "switch" and is expressed as a percentage. Finally, the isotopic space created by the sources also affects the bias. When source signatures vary over time, changes in the isotopic space can influence the bias. This is known as the "source variation effect."

### *In silico* experiments

To analyse the behaviour of bias and the influence of different factors, we designed *in silico* frameworks. Two types of systems were generated: (a) a constrained system with $n$ tracers for $n+1$ sources, consisting of two studied isotopes and three distinct food sources, and (b) an unconstrained system with more than $n+1$ sources for $n$ tracers, consisting of two studied isotopes and four food sources.

Within these two systems, four diet scenarios were examined:

1. Constrained system, single-source diet: The real diet is composed solely of source 3.

2. Constrained system, mixed diet: The real diet consists of 15% source 1, 45% source 2 and 40% source 3.

3. Unconstrained system, mixed diet: The diet consists of 20% source 1, 20% source 2, 40% source 3 and 20% source 4.

4. Unconstrained system, single-source diet: The real diet is composed solely of source 3.

For each experiment, the DMM was applied over a time window T (expressed in days) to simulate consumer signatures over time based on real diet contributions. The contribution increments were set to 0.05‰, resulting in 219 potential solutions within the constrained framework and 1,659 solutions within the unconstrained framework. Both SMM and SMMΔ were subsequently used to estimate the contributions of diet. For each model, either the top 5% of best-fitting solutions were retained, or all solutions that fit the data within a 0.5‰ tolerance were retained. Finally, bias values were calculated following the methodology outlined in Equation (7) and Equation (8) (Fig 2).

This *in silico* setup (see Fig 3) was designed to quantify the bias caused by various factors. To simplify the analysis, TDF values were set to zero for all sources and isotopes, as they do not affect the studied bias factors if kept constant. Similarly, each element's concentration ($q_s$) was set to 1 for all sources and isotopes. Across all the experiments, the combined effects of λ (isotopic turnover rate) and T (experiment duration) were examined. To account for different scenarios, each experiment was conducted with five distinct λ values (0.002, 0.0065, 0.011, 0.0155, and 0.02 $d^{-1}$) and four T values (2, 101, 200, and 300 d), resulting in 20 unique λT configurations ranging from 0.004–6 (no unit). Additionally, we verified that the size of the isotopic space (the area within the source polygon where the consumer typically evolves) did not influence the system's maximum bias (see S2 Appendix). Using this setup, two different experiments were conducted.

The first experiment quantified the diet switch effect (the effect of $\delta X(0), \delta Y(0)$ values) on bias. In this setup, source signatures were held constant over time with the following values: the source1 signatures were 10‰ for isotope 1 and 5‰ for isotope 2; the source 2 signatures were 0‰ for both isotopes; the source 3 signatures were 5‰ for isotope 1 and 10‰ for isotope 2; and the source 4 signatures were 2‰ for isotope 1 and 6‰ for isotope 2. Then, three consumer signatures were generated within the isotopic space formed by the sources (see Figs 4–5 in the Results section). Because

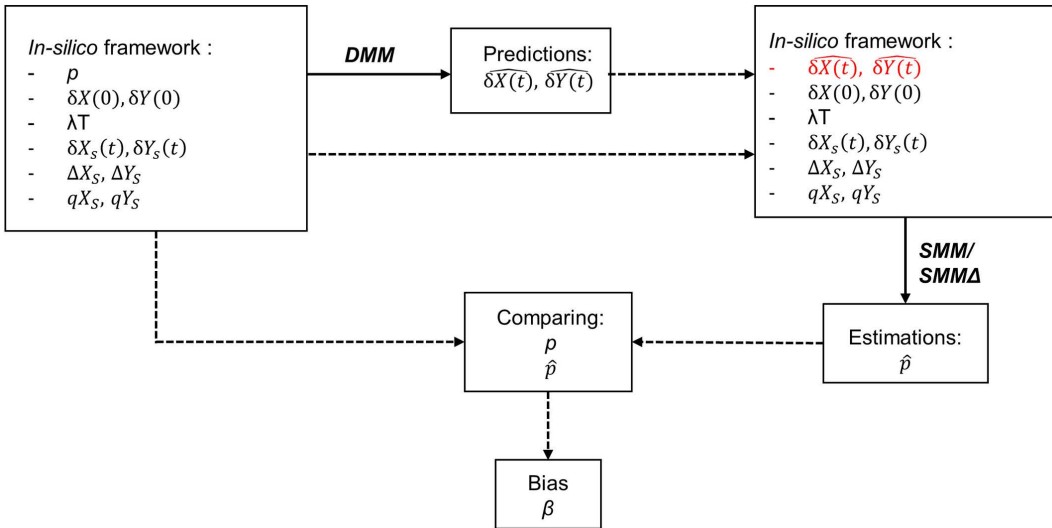

**Fig 2. *In silico* process for estimating the bias.** The solid lines represent the use of a mixing model, and the framework boxes indicate the data used in the corresponding mixing models (see Fig 1). Each box represents a step in the bias estimation process, and the dashed arrows show the data used at each step and the order in which it is applied.

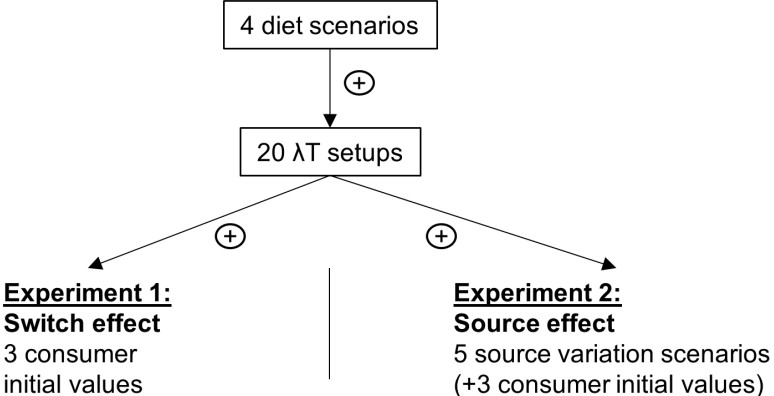

**Fig 3. *In silico* experimental setup.** Each column represents a distinct experiment, showing the effect being tested and the number of modalities used, such as the number of tested diets, the λT value tested, the number of consumer initial values and the source variation scenarios.

the source signatures remained stable over time, both the SMM and the SMMΔ yielded identical results; therefore, only the SMM results are presented. For each case, all solutions yielding estimates within a 0.5‰ tolerance were retained. Bias (%) was calculated for each initial consumer value. Additionally, the equilibrium isotopic signature ($\delta X_{eq}$) was estimated based on the actual diet and source signatures. The switch percentages were also calculated. This procedure was repeated for the three different switch diet scenarios.

The second experiment assessed the source variation effect and explored its potential relationship with the diet switch effect. Three consumer values before the switch (same as the switch effect values) were implemented to study the source variation effect under different switch conditions. Four different scenarios of source signature dynamics were

implemented, and bias was estimated for both the SMM and SMMΔ (more details are provided in the S3 Appendix). In constant-source scenarios, where exact solutions were produced, all solutions yielding estimates within a 0.5‰ tolerance were retained. In scenarios with dynamic sources, the top 5% of best-fitting solutions were retained. As in the first experiment, the models were evaluated across 20 λT configurations and four diet scenarios. The objective was to compare bias under conditions where sources remained stable over time with bias generated when sources were dynamic.

### Verifying the reliability of the DMM via an *in silico* dataset

To validate the robustness of the dynamic mixing model (DMM) under varying levels of data complexity, an *in silico* dataset was generated with a known diet over time. This approach allowed us to assess bias in DMM-based diet estimations under conditions that simulate real-world field sampling (i.e., low-frequency sampling). The goal was to evaluate the accuracy of the DMM in estimating dietary contributions when faced with incomplete or sparse data—situations that are common in field studies. The experiment employed DMM in two distinct roles: first, as a theoretical tool to create a complete, high-resolution dataset capturing the full isotopic dynamics of the system over time; second, as an applied tool to work with a subset of the data—a "field-sampled-like" dataset—representing limited and irregularly sampled consumer and source signatures, as might occur in real-world studies.

The data were generated over a 500-day period, with isotopic data collected daily. To simulate a complex environment, we modelled a system with two isotopes studied for a consumer with three potential food sources. The dietary contributions of these sources varied over time: the contribution of the first source was fixed at zero throughout the simulation, the second source followed a sinusoidal function over time (described by specific parameters in Eq. 9), and the contribution of the third source was calculated as 1−(contribution of the second source). Further details on the parameters and calculations are provided in the S4 Appendix.

$$s2\_contrib(t) = 0.5 \, sin(0.025 \, t) + 0.5 \tag{9}$$

Some source signatures were also modelled as dynamic. The first source remained constant over time, with signatures of 10‰ for the first isotope and 5‰ for the second isotope. The signatures of the two other sources were randomly generated using a Brownian motion model (see Ballutaud et al. [13]). Using these dynamic source signatures, consumer isotopic signatures over time were generated using the DMM equation (Eq. 5). For simplicity, the concentrations of the elements in all the sources were set to 1, and the TDF for every source and isotope was set to 0‰. The isotopic turnover rate (λ) typically ranges from 0.002 d$^{-1}$ to 0.2 d$^{-1}$ [50]. For this simulation, a moderate turnover rate of λ = 0.02 d$^{-1}$ was chosen. The increment (i.e., the precision of the proportions) was set to 0.05‰, a level verified to have no significant impact on the results while optimizing the computational efficiency. This setup provided a complete simulated dataset that represented the potential dynamics of an isotopic system in the environment.

The "field-sampled-like" dataset was then simulated to simulate real-world conditions. Consumer signatures were sampled on 10 distinct dates, chosen randomly without regard to seasonality or the interval between samplings. These dates were used to extract consumer isotopic signatures, forming the basis of the "field-sampled-like" dataset. Source signatures were also sampled on the same dates, with linear interpolation applied between two sampled points to approximate the source dynamics (as would be done in field data, where variations between sampling dates are unknown). The DMM was then applied using these limited data points and the same model parameters as in the environmental setup (same $q_i$, precision, TDF and λ). For each run, all solutions fitting the data within a 0.5‰ tolerance from the applied DMM were retained. To compare performance, both the SMM and SMMΔ were also run using the same data. To capture the full variability of the system, this process was repeated 100 times, creating 100 sets of 10 sampling dates. Finally, the bias of the applied DMM (using only the sampled data) was computed using Equation (8) on each kept solution, which was compared with the real mean diet over the period between two samplings.

## Case study

To demonstrate the application of the developed MMs, particularly the DMM, we used a dataset from Inger et al. [33]. This dataset describes the carbon and nitrogen isotopic signatures of geese in Northern Ireland sampled nine times between October 2003 and April 2005. Isotopic measurements were conducted on two different tissues, plasma and blood cells, each with distinct isotopic turnover rates. Compared with blood cells ($\lambda = 0.03$ d$^{-1}$, corresponding to a half-life of 20 days), plasma cells exhibit a much faster turnover rate ($\lambda = 0.33$ d$^{-1}$, corresponding to a half-life of 2 days) [33,51]. For this study, we focused on blood cell data, and the results for plasma tissue are detailed in the S5 Appendix. The dataset also includes constant isotopic signatures for four potential dietary sources: *Zostera* spp., *Ulva lactuca*, *Enteromorpha* spp. and terrestrial grass. We used $\Delta^{13}C = 1.63$‰ and $\Delta^{15}N = 3.54$‰ for TDFs [33]. Further details of the dataset and additional data illustrations can be found in the S5 Appendix.

We applied both the SMM and the DMM to this dataset to estimate the contributions of each source to the geese's diet at or between each sampling period. Because the dataset does not report temporal variation in source signatures, averaging the sources over a time period results in identical outcomes for both the SMM and SMMΔ models. The model was set to retain the top 1% of the best solutions (50 solutions) with a generated precision increment of 0.01‰. To address the high correlation between the two dietary sources *Ulva lactuca* and *Enteromorpha* spp., we combined them into a single source (*Ulva* x *Enteromorpha*), as recommended by Phillips et al. [22]. For each sampling period, we calculated the λT value (the time in days between two samplings multiplied by the turnover rate). We hypothesized that the DMM results would closely approximate the geese's actual diet and used these results as a reference to evaluate potential biases in the SMM estimates.

## Results

### *In silico* experiments

The initial isotopic value of the consumer prior to a diet switch had a significant influence on the bias observed in the SMM (Fig 4). At low λT values, the bias increased with the percentage of switch. Specifically, at very low λT values, the median bias was observed to be 0% for initial values close to equilibrium, increasing to 50% for moderately distant initial values, and peaking at 100% for the most distant initial values. This trend highlights the pronounced effect of λT: as λT increased, the bias steadily decreased. Once the λT values exceeded 4, the bias decreased to zero, effectively eliminating the impact of the diet switch effect on the SMM. This pattern was observed for both constrained and unconstrained systems, although the uncertainty around the median values was greater for the unconstrained cases.

When the reference diets were compared, it became clear that the maximum bias caused by the switch effect was influenced by the switch percentage (distance of the initial consumer signature from equilibrium relative to the isotopic space). In Fig 5, where the maximum distances to equilibrium were 40% and 60% of the switch, the maximum median bias reached only 80%, whereas in Fig 4, the bias increased to 100%. In the second diet scenario (Fig 5A), a switch effect was still observed. Initial values with low distances from equilibrium presented biases of up to 20%, whereas higher distances resulted in biases reaching 80%. Similarly, when λT exceeded 4, no bias was generated by the consumer's signature before the switch. In the third diet scenario (Fig 5B), the system was unconstrained, involving four food sources for two isotopes. A similar pattern was observed, but even at high λT, the median bias remained at approximately 20% rather than dropping to 0%. Additionally, the unconstrained case exhibited greater uncertainty in the bias, especially at high λT values. In constrained cases, when λT > 4, both the median and quartiles equalled 0%, whereas in unconstrained cases, the median bias was 20%, with quartile values ranging between 10% and 40%.

The use of the SMM when food source signatures vary over time introduces significant bias (Fig 6). Specifically, under dynamic source conditions, the median bias in the SMM could reach 100% under low λT conditions, whereas it remained at 0% with the same initial value under constant source conditions. This additional median bias gradually diminished as λT

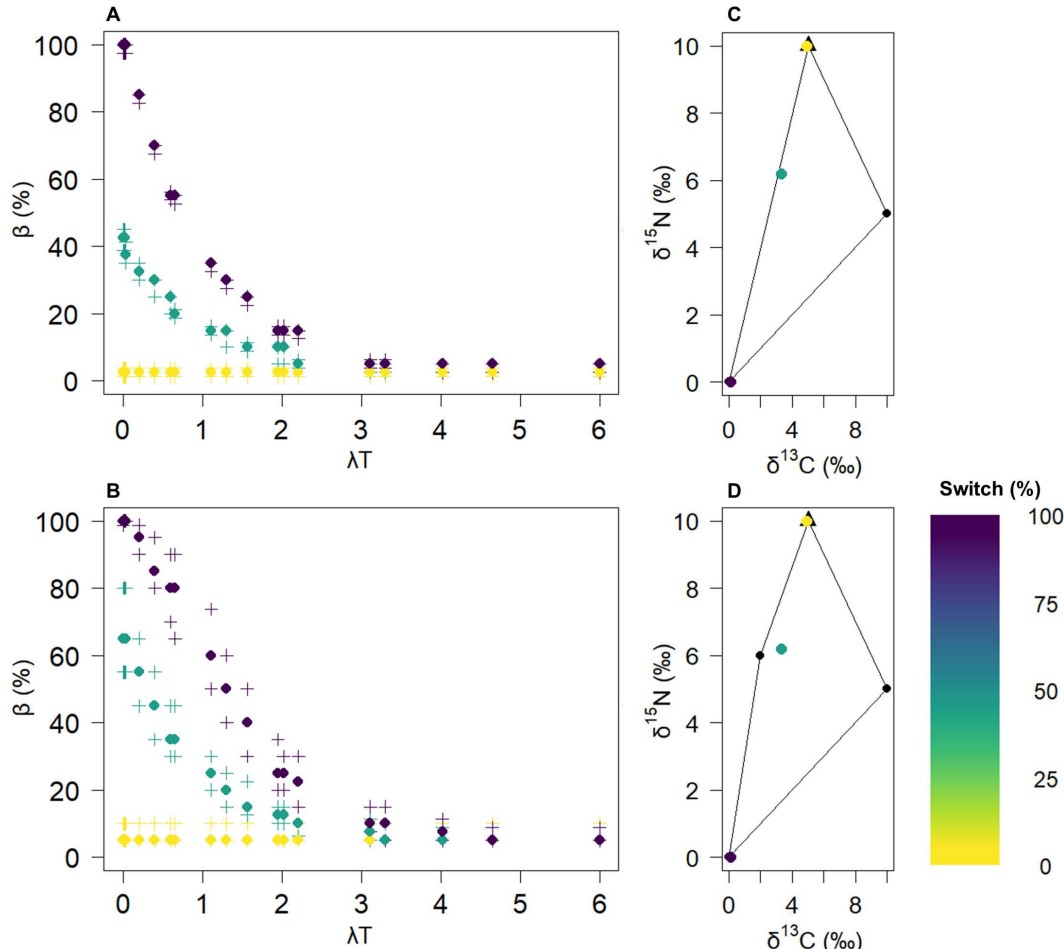

**Fig 4. Diet switch effect on bias for a single-source diet for constrained (diet 1, 100% of source 3) and unconstrained systems (diet 4, 100% of source 3).** The first row shows the results for diet 1 (A and C), and the second row shows the results for diet 4 (B and D). The dots in Panels A and B represent the median bias values for all the solutions within a 0.5‰ tolerance. The crosses represent the quartiles. Each colour corresponds to an initial value and its distance from equilibrium (switch percentage). These are shown in the source polygons in Panels C and D, where the triangle indicates the equilibrium signature.

increased. In comparison, SMMΔ demonstrated a significantly reduced additional bias under similar conditions. Although the difference in additional bias between SMM and SMMΔ was less pronounced in unconstrained systems, it remained significant (see S6 Appendix).

### Reliability of the DMM via an *in silico* dataset

The *in silico* environment produced a complex and dynamic system for studying bias (Fig 7). As shown in Fig 8, for λT > 2, the median bias values covered a wide range, spanning from 10–50% across all the models. For all three models, the bias increased when λT > 2, a trend not observed in the previous *in silico* experiments (see Figs 4–6). This can be attributed to the presence of multiple diet switches in this experiment, compared with the single diet switch explored in earlier setups, as well as the variable sources signatures over time. A significant difference in bias between the models was observed in the λT < 2 region. For the DMM (Fig 8A), the bias remained at 10%. In contrast, the static mixing models exhibited a distinct pattern of greater bias (50%), which decreased with increasing λT, which was consistent with earlier observations. At

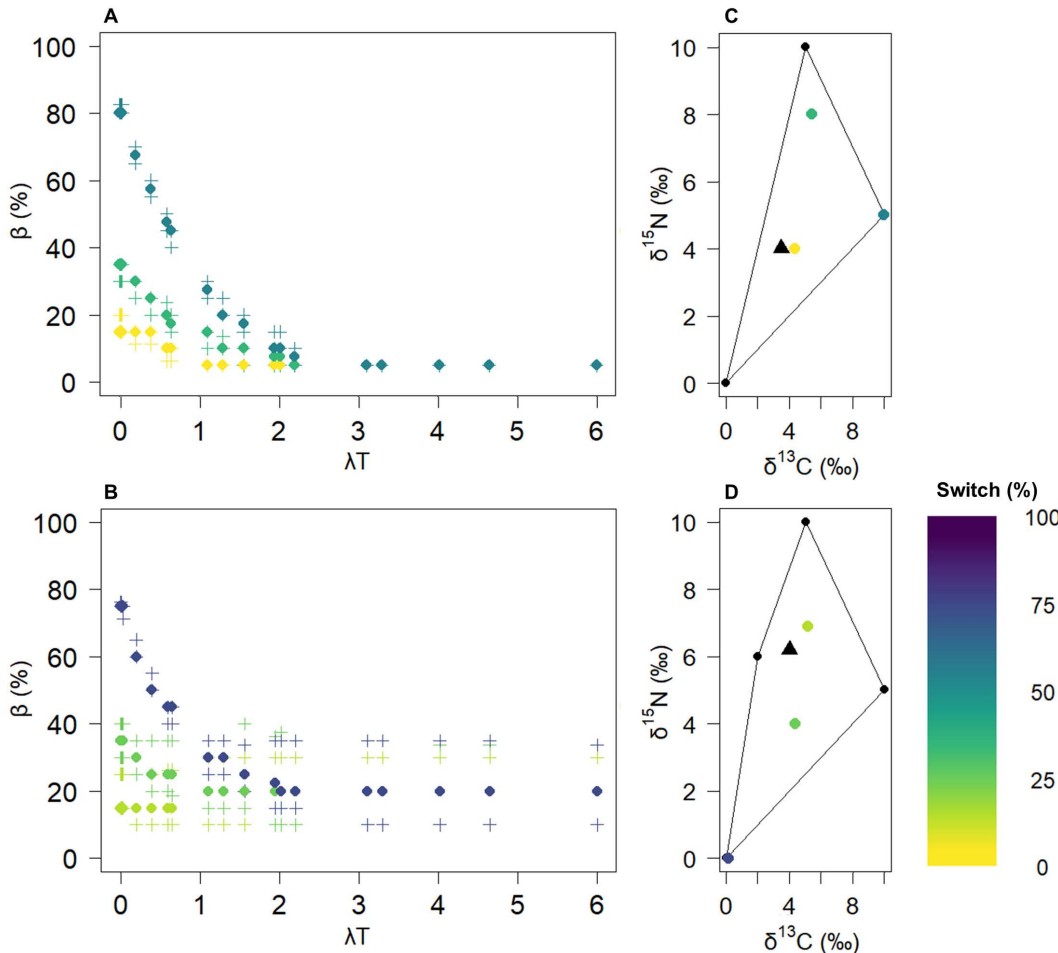

**Fig 5. Effects of diet switching on bias for mixed diets for constrained (diet 2, 15% of source 1, 45% of source 2 and 40% of source 3) and unconstrained (diet 3, 20% of source 1, 20% of source 2, 40% of source 3 and 20% of source 4) systems.** The first row shows the results for diet 2 (A and C), and the second row shows the results for diet 3 (B and D). The dots in Panels A and B represent the median bias values for all the solutions within a 0.5‰ tolerance. The crosses represent the quartiles. Each colour corresponds to an initial value and its distance from equilibrium (switch percentage). These are shown in the source polygons in Panels C and D, where the triangle indicates the equilibrium signature.

very low λT values and in general, the DMM performed significantly better than the two static mixing models did in terms of diet estimation. A difference can also be observed between SMMΔ (Fig 8B) and SMM (Fig 8C) when λT is between 1 and 2. In this zone, the median bias decreases more rapidly for SMMΔ than for SMM, indicating better performance of the integrated model.

## Case study

Notable differences emerged between the results of the two models (SMM and DMM; Fig 9A, B). The first distinction lies in their approach to diet estimation. The SMM estimates the diet at each individual sampling point, whereas the DMM operates over time windows, requiring data from at least two sampling points to estimate the diet. For example, the estimation at 92 days reflected the mean diet from Day 0 to Day 92. Second, differences appeared in the estimated contributions of each source. For example, on Day 123, the SMM estimated a diet primarily composed of Grass and *Ulva* and *Enteromorpha*, whereas the DMM estimated a diet including Grass, *Zostera*, *Ulva*, and *Enteromorpha*, with a majority of

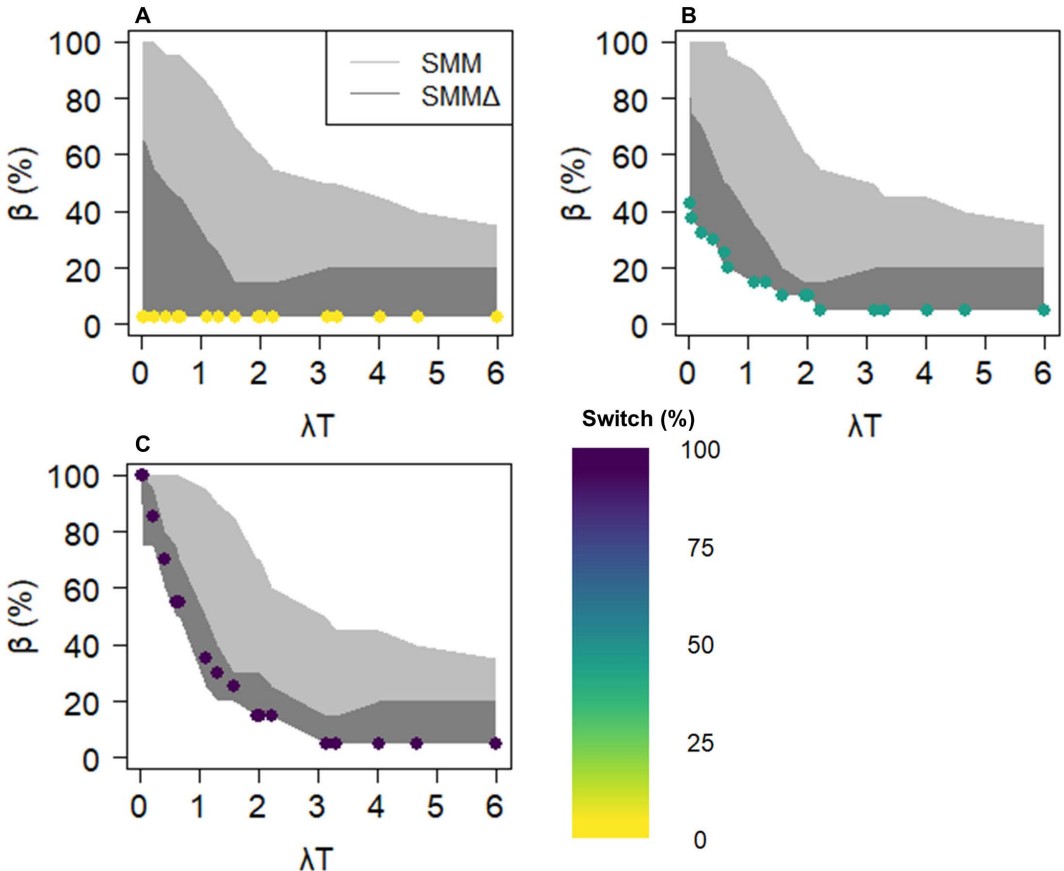

**Fig 6. Effect of source variability on bias as a function of λT for three initial consumer values in diet 1 (with three sources, 100% of the diet from source 3).** The median biases are represented for three consumer initial values: one close to equilibrium (A, 0% switch), one in the middle of the polygon (B, 50% switch), and one far from equilibrium (C, 100% switch). For more details, see Fig 4C. The coloured points represent the median bias value in the source-constant scenario for solutions falling within a 0.5‰ tolerance. The grey envelopes represent the maximum and minimum median biases in the dynamic source scenarios for the top 5% of the best solutions. Quartiles ranges are not shown here for the sake of clarity but can be observed in Fig 4A.

Grass. Visually, the period with the greatest difference occurred between Days 400 and 500. These disparities were also reflected in the estimated median bias (Fig 9C). The two highest bias estimates were observed at the sampling points on Days 457 and 123. The estimated median bias ranged from 0–35% across λT values ranging from 1.11–9.7. Here, the λT value is derived from the period between two sampling points, varying from 31 days (λT = 1.11) to 273 days (λT = 9.75), assuming a constant λ throughout the study. Consistent with theoretical predictions under constant isotopic source values for both pattern and observed values, the estimated bias decreased as λT increased, becoming negligible when λT > 4.

## Discussion

This study provides several key insights into the sources of bias in mixing models (MMs), particularly stemming from the underutilization of isotopic dynamics. First, we demonstrate that bias is strongly influenced by the λT product, underscoring the critical interplay between time variability and isotopic integration—factors that are often overlooked in the literature. By incorporating metrics such as λT, we not only generalize *in silico* results but also facilitate their direct application to fieldwork, as both λ and T can be estimated in the field. This bridges the gap between the theoretical bias in MMs and the

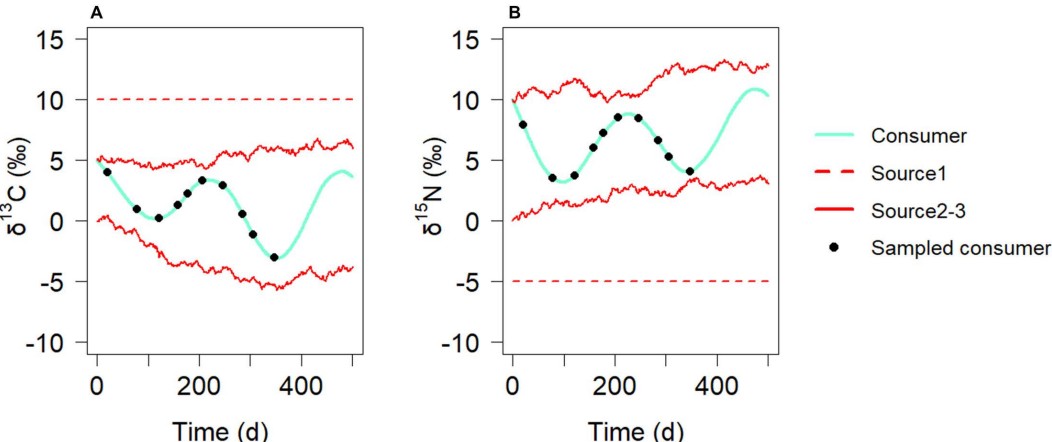

**Fig 7. Carbon (A) and nitrogen (B) simulated signatures over time in the *in silico* environment and an example of 10 sampled consumer signatures.** The blue line represents the simulated complete time series of the consumer's isotopic signatures over time. The red lines are the simulated complete time series of the sources' isotopic signatures over time. The black dots are examples of 10 sampled values used for diet 1 simulation.

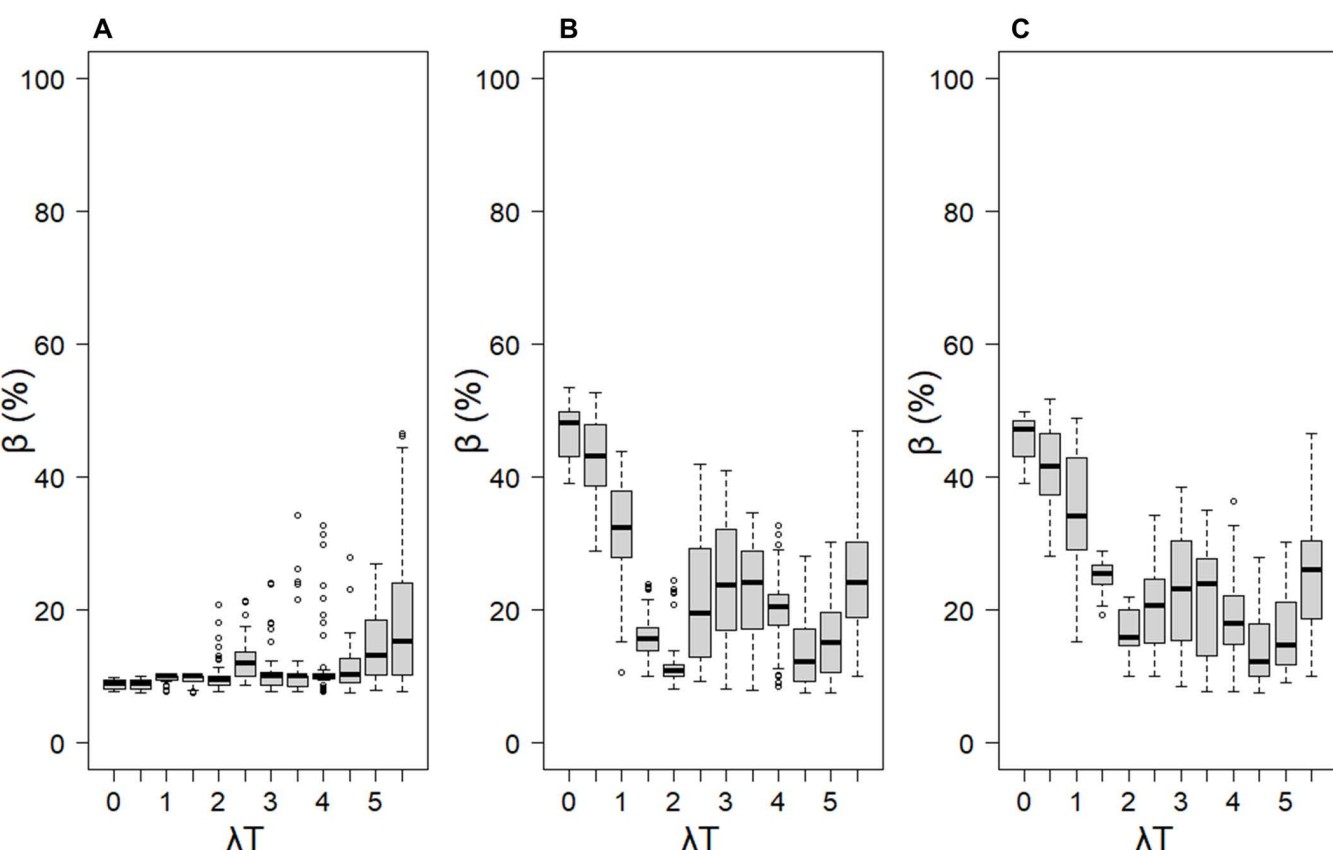

**Fig 8. Boxplot of median biases for the three models relative to the field-sampled-like dataset.** (A) Applied DMM, (B) SMMΔ, and (C) SMM median bias using the field-sampled dataset. The observations were separated into 10 classes of λT (size 0.5). Quartile ranges are not shown here for the sake of clarity since they may widen the bias magnitude.

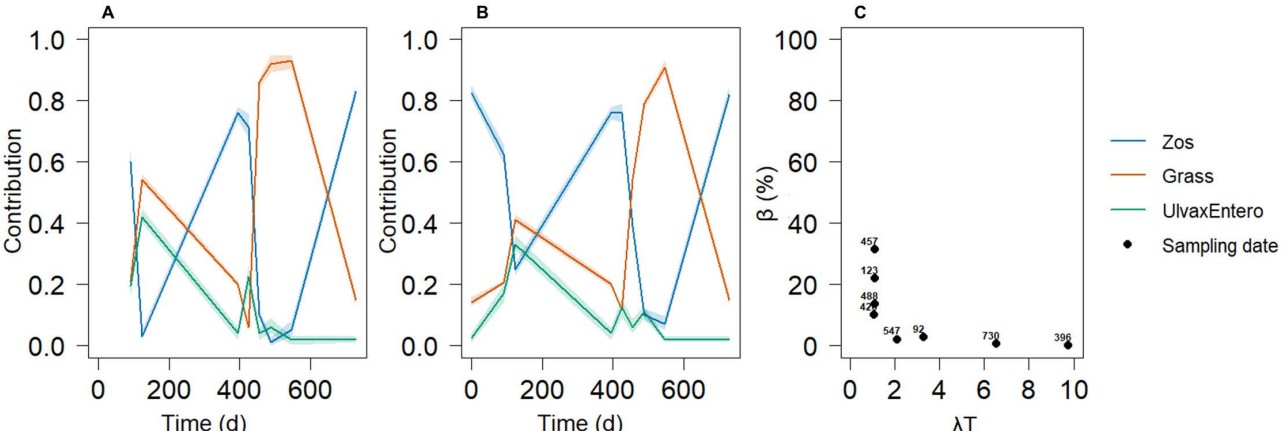

**Fig 9. Comparison of source contribution estimations over time by the SMM and DMM in the geese dataset and associated bias estimations.**
Panels (A) and (B) depict the estimated contributions of each food source over time, for the SMM and DMM, respectively. The lines represent the median estimates, and the envelopes represent the quartiles. Panel (C) displays the median bias according to λT, with the median DMM estimations used as the reference diet. Quartile ranges are not shown here for the sake of clarity since they may widen the bias magnitude.

errors of estimation observed in real-world datasets. Another important contributor to bias is the switch effect, which arises from the consumer's distance from equilibrium at the time of a diet switch. This can lead to substantial biases, particularly when source signatures vary over time, further exacerbating the issue. In addition, we assessed the recommendations of Phillips et al. [22], confirming their effectiveness in reducing bias in static mixing models (SMMs) and providing additional guidelines for their optimal application. Furthermore, we extended the accessibility of the DMM, allowing for a more comprehensive comparison with other MMs. This comparison highlighted both the strengths and areas for improvement in the DMM, demonstrating its robustness and reliability in accurately estimating dietary contributions.

## Comparison between the DMM and SMMs

The developed DMM is applicable to any system, regardless of the number of studied isotopes or food sources, significantly broadening its applicability to diverse datasets compared to Ballutaud et al. [13]. Its primary advantage lies in its ability to account for dynamics in both source and consumer isotopic signatures. Under typical conditions for mixing model applications (λT<2, representing sampling over fewer than 200 days for λ=0.02 d$^{-1}$), the DMM exhibits minimal bias, whereas static models become completely biased, making the DMM the most reliable deterministic mixing model available. The *in silico* environment used in this analysis demonstrates the robustness of the DMM. It represents a highly complex isotopic system, mimicking the conditions of a stable isotope study where sources, consumer signatures, and the diet itself vary dynamically over time. Despite this complexity, the DMM consistently produces diet estimations with minimal bias at low λT values (λT<2), unlike those produced by static models (without accounting for data uncertainty). This capability highlights the ability of the DMM to manage isotopic temporal dynamics effectively, making it particularly suitable for cases where the λT values are close to 0, which often poses challenges for static models.

Additional recommendations for selecting and applying mixing models regardless of whether deterministic or Bayesian approaches are used, are presented in Fig 10. First, it is crucial to assess whether the isotopic signatures of sources and consumers remain stable over time. If not, the use of the SMM should be avoided. When only source dynamics are present, the SMMΔ can be used, provided that the sources are sampled over an adequate λT value. When the consumer's isotopic behaviour is dynamic, the DMM should be employed to avoid bias. At high λT values in complex and dynamic systems, all three models show high variability in bias, which can be partially attributed to

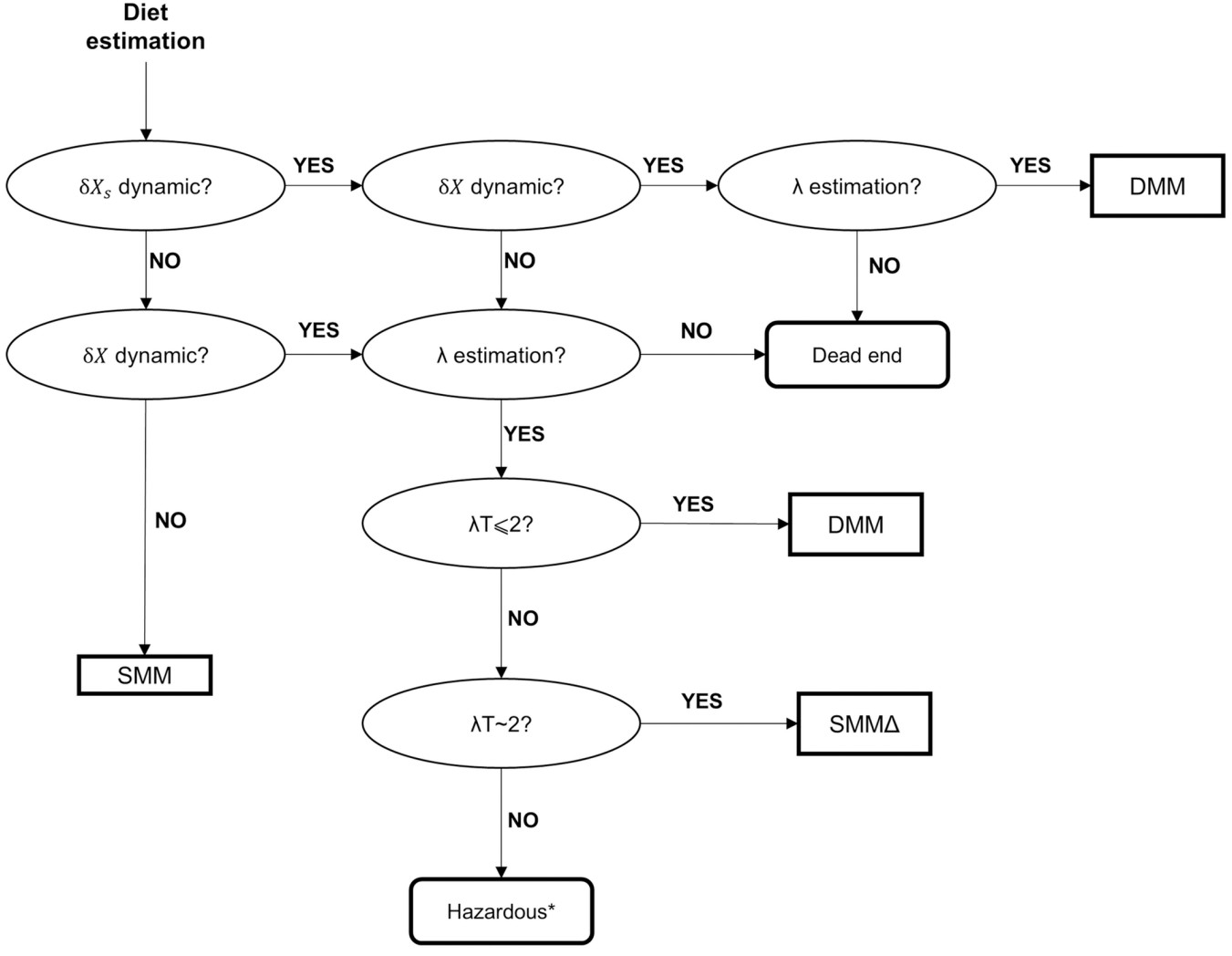

**Fig 10. Decision tree for selecting the most suitable model among the SMM, SMMΔ, and DMM.** Adapted from Ballutaud et al. [13]. Each circle represents a key aspect of the system that should be considered when selecting a mixing model, guiding the choice among the three models. *The use of any MM in the λT>2 condition may lead to a high bias depending on the system and its dynamics.

the complexity of the system itself. This variability may also stem from temporal patterns in diets, such as the sinusoidal shifts in source contributions and consumer signatures observed in this study. Phenomena such as phase opposition and phase shifts can cause exceptional bias, even with identical λT values, regardless of the model. Future studies could explore alternative patterns of diet variation, such as consistent periodic shifts or random dietary changes.

Ultimately, λT values greater than 2 are unsuitable for accurate diet estimation, regardless of the model applied. The sampling time should be adjusted according to the system's isotopic dynamics, ensuring that the time window reflects the isotopic incorporation process but does not extend too far to compromise the relevance of diet estimation. Estimating a diet with λT>2 is risky for any model (SMMΔ or DMM), and if SMMΔ is used, the optimal sampling time corresponds to λT~2 (see Fig 10).

## Importance of considering isotopic turnover and sampling/experiment time (λT) in isotopic studies

Errors in estimating source contributions using isotopic mixing models are heavily influenced by the system's parameters, with λT emerging as the most critical factor driving bias. Under similar conditions, bias can range from 0–100%, depending on the λT values, underscoring the importance of accurately estimating turnover rates in isotopic integration and mixing model applications [10,21]. Specifically, organisms with low isotopic turnover which were sampled over short study durations (λT < 1 in this study), exhibited the highest bias in static mixing models. A key recommendation for applying static mixing models is to manage the λT values carefully to minimize bias. The bias is highest when λT approaches 0. In our *in silico* experiments, which involved a single diet switch, λT values exceeding 4 consistently yielded unbiased estimates. However, achieving such values may require long sampling periods [33], which can be logistically challenging. Moreover, these experiments modelled systems with only one diet change and a stable diet otherwise. In systems with dynamic diets, the lowest bias in static models typically occurs at λT ≈ 2.

However, extended sampling periods introduce the risk of uncorrelated data. For example, annual measurements of consumer signatures may fail if diets shift multiple times within that period or if migration occurs. In such scenarios, mixing models may fail to detect or account for diet changes, leading to bias. In systems with high complexity—characterized by multiple diet shifts and fluctuating sources over extended periods (λT > 4)—all models, including dynamic ones, may exhibit increased bias. Ballutaud et al. [13] suggested that the optimal integration time for incorporating sources corresponds to twice the half-life ($\frac{2\log(2)}{\lambda}$), resulting in a λT value of 1.38. To estimate a minimum time window for future source sampling while using an SMMΔ, the following expression can be applied: $T = \frac{1.38}{\lambda}$. This value also represents the λT zone where SMMΔ performs better than SMM.

The λ parameter is critical for understanding bias in mixing models, yet accurately estimating it can be challenging. Various methods exist to estimate isotopic turnover, including diet-switch experiments in controlled or semi-controlled environments using the time model (Eq. 3) [52–54] or the mass model [40]. Alternatively, statistical models based on experimental data can estimate λ by incorporating factors such as body mass and temperature [50]. Physiological models, such as the dynamic energy budget (DEB) model, offer another approach [32,55], requiring variables such as the weight, temperature and food availability over time. However, obtaining these estimates is often complex and not universally feasible, especially for species with limited physiological data, as λ is species-specific [56]. Determining λ for *in situ* data is even more difficult because of its dynamic nature, which is subject to seasonal variations [32] and dependence on individual body mass [12,54]. Despite these challenges, neglecting isotopic turnover rates when static mixing models are used can lead to significant errors in diet estimation.

Although it is impossible to directly control the λ value in a studied system, the experiment duration (T) can be adjusted through the sampling protocol to mitigate potential bias in static mixing models. Aligning the T value with the λ value can partially manage bias. This requires temporal sampling with at least two distinct sampling dates. For example, when individuals with very low turnover rates (e.g., oysters during winter [32]) are being studied, extending the sampling period can minimize bias by increasing the λT value. Conversely, for organisms with high turnover rates (e.g., fish larvae or copepods [52,57]), shorter sampling intervals can result in low-bias estimations.

## Mixing models and bias in the field

Conducting temporal isotopic sampling can be challenging, particularly in remote areas such as deep seas [58,59] or polar regions [60]. However, temporal aspects can often be considered in many study systems without requiring extensive additional sampling efforts. A review of 103 articles (see S1 Appendix) revealed that only 18% of the studies accounted for temporal effects. However, 54% presented datasets with multiple consumer signature measurements over time, and 43% included multiple source signature measurements. This suggests that at least 25% of these studies could theoretically incorporate temporal aspects in mixing models using their existing datasets. By accurately estimating the sampling period and calculating λ, these studies could assess the bias in their dietary estimations and evaluate the need for a dynamic

mixing model (DMM) (see Fig 10). The applicability of the DMM depends on the type of dataset available. One scenario involves studies where multiple source and consumer signatures are already collected over time. This is often the case in studies investigating seasonal effects, where one signature is sampled per season [29,61], or in aquaculture studies tracking diet changes in growing animals with one sampling per growth period [62]. Some studies also sample prey signatures over a period related to λ before measuring consumer signatures [28,63–65]. These studies, which already account for temporal dynamics, could readily adopt a DMM for more accurate estimations.

Another scenario involves studies that initially treat their data as static but, in fact, include multiple sampling points over time. These studies could transition to a temporal framework with minimal adjustments. The most straightforward case occurs when repeated sampling is conducted over time for both consumer and source signatures [66–69]. These datasets can directly support temporal analysis. In other cases, where only one of either the consumer or source signatures is sampled over time, the DMM can still be applied with minimal additional data. For example, Johnson et al. [70] presented multiple consumer signatures over time and attempted to estimate dietary changes with only a single estimation of source signatures. By adding a second sampling of source signatures, the study could complete a temporal survey. Similarly, studies employing opportunistic sampling, such as those by Gama et al. [71], Ogilvy et al. [72], and Rosas-Luis et al. [73], could benefit from a temporal approach, as they incorporate long-term consumer data. Although uncertainties might arise owing to inconsistent sample sizes in opportunistic studies, the DMM can help estimate diets by inferring source and consumer signatures over time.

This study evaluates the effectiveness of the recommendations given by Phillips et al. [22] in mitigating bias in static isotopic mixing models. For bias caused by source variability over time in constrained cases, following Phillips et al.'s recommendations and using SMMΔ instead of the standard SMM significantly reduces bias. However, when addressing the diet switch effect, both static models fail to account for variability in consumer isotopic signatures. This issue is particularly relevant to Phillips et al.'s recommendation that consumer signatures should remain within the range of the source signatures. If this condition is not met, it suggests that the source polygon shape has shifted and that the system—especially the consumer—is not at equilibrium. Under such circumstances, a static mixing model assuming equilibrium [8] should not be applied.

These recommendations aim to minimize bias in isotopic field applications, but adherence to these guidelines remains inconsistent across studies. In our analysis of 103 articles (see S1 Appendix), 40% cited Phillips et al., indicating widespread awareness of these recommendations within the isotopic research community. However, adherence to these guidelines varied considerably. Specifically, we selected three key guidelines for evaluation: (1) whether the authors carefully selected trophic discrimination factors (TDFs), (2) whether they ensured that their consumer's isotopic signature fell within the source polygon, and (3) whether temporal sampling for the sources was conducted. We found that 5% of the articles followed none of these guidelines, 58% followed only one (usually the choice of TDF), 34% followed two (typically TDF selection and checking the source polygon), and only 6% followed all three guidelines. Furthermore, as previously noted, only 26% of the articles discussed λ, a crucial parameter for minimizing bias. This limited incorporation of temporal considerations is concerning, as these factors can significantly affect the reliability of dietary estimations. As demonstrated in this study, neglecting these factors can introduce substantial bias, compromising the accuracy of the results. This underscores the need for broader application of Phillips et al.'s recommendations to ensure more reliable and accurate isotopic analysis in future studies.

## DMM limitations

This study explored several factors influencing bias, such as source and consumer variability and the λT value, as these parameters directly affect bias. All these factors can be managed using the dynamic mixing model (DMM). However, some sources of bias are inherent to mixing models (MMs) and cannot be corrected by the DMM. For example, the DMM, like any MM, cannot distinguish between different sources with very similar isotopic signatures or address the imprecision in

diet estimation when studying an unconstrained system. Additionally, MMs provide only an average diet over the time window under study. When applied over long time windows (depending on the associated λ value, e.g., with annual sampling for large species or monthly sampling for larvae with high λ [52]), this can result in reduced precision and potential bias.

The trophic discrimination factor (TDF) also significantly impacts diet estimation [10,22]. When analyzing isotopic dynamics, it is essential to account for TDF, as it can vary with food quality and physiology [15,56,74]. A diet switch, for example, can lead to changes in fractionation. Moreover, TDF is influenced by physiological factors, increasing under nutritional stress and disease and decreasing during growth [12,75]. As such, the TDF varies alongside the turnover value throughout an individual's life [12], making it a dynamic parameter. Although this study did not focus on TDF values or their dynamics, which could introduce an additional layer of bias beyond what was examined here, it is important to note that this would create an even more complex system where identifying bias becomes more challenging. The code used for modelling the DMM was developed with the concept of a dynamic TDF in mind. Users have the option to apply the DMM with a time-varying TDF to account for this additional complexity.

Another limitation of the DMM as implemented (using a frequentist approach) lies in its reliability assessment. This model assumes no data or parameter-based uncertainties, making it impossible to investigate error when applied to a dataset. Moreover, it does not assess the correlation structure within the model itself unlike standard mixing models [19]. As a result, the DMM is less user-friendly for now, providing no direct estimates of contribution correlation and yielding lower precision in error structure estimation compared with existing mixing models. These issues could be managed by adapting the DMM to a Bayesian approach.

## Conclusion and further perspectives

Bias is a valuable tool for identifying and anticipating estimation errors in static mixing models. In this study, we identified factors that lead to biased estimates and quantified their impact. This serves two main purposes for readers. First, it alerts researchers to the use of mixing models to estimate the potential for significant bias in diets and underscores the importance of accounting for it. Second, it helps them recognize common scenarios that lead to high bias, enabling them to address these issues in their datasets, sampling plans, and analyses. Bias studies such as this one may reveal additional factors not addressed here. For example, running the DMM on diet-switch experimental datasets [52,53,76] could help quantify bias and provide concrete examples of specific scenarios and their associated biases. Such case studies could offer insights for predicting bias in other systems, helping researchers avoid situations where estimation errors are particularly high.

The DMM implemented in this study, as well as in Ballutaud et al. [13], uses techniques derived from the time model [40,47] to estimate isotopic dynamics. However, other modelling approaches, such as the mass model [8] or a dynamic energy budget (DEB) model [55], could also be adapted to incorporate temporal dynamics. Transitioning from a static to a time–dynamic isotopic model involves incorporating λ and considering $\delta X_{eq}$ as a dynamic parameter. The DMM can also be applied to estimate an individual's diet throughout its lifetime. This requires analysing tissues that preserve the consumer's isotopic signature over time, including hard tissues such as otoliths [77] or baleens [78], and soft tissues such as fish eye lenses [79,80]. It also necessitates temporal monitoring of potential food sources that may change over time. These datasets can be integrated with isoscape and mobility models to estimate migrations, as demonstrated by Trueman et al. [78], who added spatial patterns alongside temporal patterns. Although this study focused primarily on time-induced dynamics—an aspect often overlooked in the literature—it is important to note that isotopic signatures also exhibit spatial variability [81,82], adding another layer of complexity to isotopic systems that are characterized by spatiotemporal variability and multiple potential sources of bias.

To further advance the DMM, adapting it to a Bayesian framework could be considered. Bayesian approaches are capable of incorporating data uncertainty [83], which is lacking from the DMM at the moment, and are increasingly used in diet estimation, as exemplified by the MixSIAR package [19]. However, the DMM cannot be directly integrated into

existing models such as SIAR [18], MIXSIAR, or FRAME [41], as these are static models that rely on the static mixing equation (Eq. 1). Although FRAME allows for alternative mixing equations, it does not account for temporal variation in sources and consumer signatures. Some attempts have been made to incorporate temporal effects in MixSIAR by adding temporal variability in source signatures as a residual error factor [42], making it effectively an SMMΔ. Alternatively, the model can be run separately for each sampling period, creating a series of SMMs. However, neither of these approaches captures the temporal dynamics of the consumer (see S7 Appendix for further details). This raises the possibility of a Bayesian DMM implementation. The existing Bayesian mixing models coded in R use the JAGS language to run Markov Chain Monte Carlo (MCMC) simulations, but JAGS does not support differential equation resolution, which is essential for the DMM (Eq. 5). A possible alternative is to implement MCMC using the Stan language, which performs similarly to JAGS while supporting differential equations.

A Bayesian DMM would offer the same diet estimation performance as the current DMM but with significant improvements in handling uncertainty. This approach allows residual error modelling without involving temporal patterns and accounts for errors related to model assumptions such as predation mode, which may require alternative mixing equations [41,42]. The novelty of a Bayesian DMM over traditional Bayesian mixing models lies in its temporal approach, which enables the integration of isotopic data. This would introduce a new parameter: $\lambda$. Furthermore, Bayesian implementation would provide flexibility in defining $\lambda$ values, allowing for the use of informative or noninformative priors based on available data quality, ultimately improving parameter estimation and uncertainty assessment.

## Supporting information

**S1 Appendix. Literature review regarding the use of mixing models to estimate diet and temporal effects.**
(DOCX)

**S2 Appendix. Isotopic space effect on maximum bias.**
(DOCX)

**S3 Appendix. Source dynamics in *in silico* experiments.**
(DOCX)

**S4 Appendix. Applied dynamic mixing model diet contributions over time.**
(DOCX)

**S5 Appendix. Dataset from Inger et al., 2006 and mixing model outputs.**
(DOCX)

**S6 Appendix. Additional results from the *in silico* experiments.**
(DOCX)

**S7 Appendix. Model comparison between the dynamic mixing model and MixSIAR.**
(DOCX)

## Acknowledgments

The authors would like to thank the SFR Campus de la Mer for its support. Additional supports were provided by the the European Union (ERDF), the French government, the region Hauts-de-France and Ifremer in the framework of the project CPER IDEAL (2021−2027) and the French government as part of the Programme Investissement d'Avenir (I-SITE ULNE/ANR-16-IDEX-0004 ULNE) managed by the Agence Nationale de la Recherche (ANR) and by the Métropole Européenne de Lille in the framework of the ISIT-U project.

## Author contributions

**Conceptualization:** Emilie Cathelin, Sebastien Lefebvre, Carolina Giraldo.

**Data curation:** Emilie Cathelin.

**Formal analysis:** Emilie Cathelin.

**Funding acquisition:** Sebastien Lefebvre.

**Investigation:** Emilie Cathelin, Sebastien Lefebvre, Carolina Giraldo.

**Methodology:** Emilie Cathelin, Sebastien Lefebvre, Carolina Giraldo.

**Project administration:** Emilie Cathelin, Sebastien Lefebvre, Carolina Giraldo.

**Resources:** Sebastien Lefebvre.

**Software:** Emilie Cathelin.

**Supervision:** Sebastien Lefebvre, Carolina Giraldo.

**Validation:** Emilie Cathelin, Sebastien Lefebvre, Carolina Giraldo.

**Visualization:** Emilie Cathelin, Sebastien Lefebvre, Carolina Giraldo.

**Writing – original draft:** Emilie Cathelin, Sebastien Lefebvre, Carolina Giraldo.

**Writing – review & editing:** Emilie Cathelin, Sebastien Lefebvre, Carolina Giraldo.

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
