## [Decision Letter · Decision Letter 0]

10 Feb 2025

PONE-D-24-58834From Static to Dynamic: Embracing dynamics in Isotopic Diet EstimationPLOS ONE

Dear Dr. Cathelin,

Thank you for submitting your manuscript to PLOS ONE. After careful consideration, we feel that it has merit but does not fully meet PLOS ONE’s publication criteria as it currently stands. Therefore, we invite you to submit a revised version of the manuscript that addresses the points raised during the review process.

The reviewers have identified significant methodological weaknesses, particularly concerning the calculation of bias and the omission of correlations in mixing models. A thorough revision addressing these fundamental issues, along with substantial improvements in data interpretation, manuscript structure, and language clarity, is required for the study to be reconsidered for publication.

We look forward to receiving your revised manuscript.

Kind regards,

Przemysław Mroczek, Dr. hab.

Academic Editor

PLOS ONE

“This work was financed by the PhD fellowship (EC) granted by the Region Hauts-de-France and the Graduate School EDSMRE from University of Lille.”

Additional Editor Comments:

**Dear Authors,**

After a thorough evaluation of your manuscript *"From Static to Dynamic: Embracing dynamics in Isotopic Diet Estimation"* , I must inform you that while the study addresses a relevant issue within stable isotope mixing model research, it requires **substantial revisions** before it can be considered for publication. The manuscript has been subjected to rigorous peer review, and both reviewers have provided detailed critiques highlighting **critical methodological flaws, issues with data interpretation, and deficiencies in manuscript clarity and structure** . To be considered for further review, the manuscript must undergo a **comprehensive revision that fully addresses all major concerns** .

Failure to address these methodological concerns will render the manuscript unsuitable for further consideration.

The **most critical concern** pertains to the **methodology** , specifically the calculation of bias and the implementation of mixing models. The definition of bias used in the manuscript is fundamentally flawed, as it results in **a trivial summation to zero** , potentially masking meaningful deviations in the data. The revised version must instead adopt an approach that assesses **individual source biases** , allowing for a clearer and statistically valid interpretation of model performance. Additionally, the manuscript **fails to account for probability distributions** , which is crucial when discussing bias in an underdetermined system. The revised analysis **must incorporate these distributions** , rather than relying solely on central values such as means or medians.

An **equally pressing issue** concerns the **implementation of mixing models** . The current brute-force enumeration disregards the **inherent correlations between sources** , significantly weakening the validity of the findings. Given that Bayesian models are the **established standard** for handling such complexities in stable isotope research, your manuscript **must integrate Bayesian methods** , either through direct implementation or via a comparative analysis that explicitly demonstrates the limitations of brute-force approaches. The revised manuscript must include a clear discussion of **MixSIAR (Stock et al., 2018) and FRAME (Lewicki et al., 2022) models** , as well as an improved justification for the chosen methodological framework.

Beyond these methodological issues, the **structure of the manuscript requires substantial reorganisation** . The discussion is overly long and repetitive, often restating results rather than providing a concise synthesis of their implications. The revised version must **eliminate redundant sections, improve coherence, and ensure that findings are positioned within the broader context of stable isotope research** . Additionally, the introduction **overgeneralises prior work** , suggesting that most studies fail to account for temporal dynamics. This statement is **too strong and must be rephrased** , as it currently lacks sufficient justification and risks misrepresenting the field.

The **interpretation of results** also requires significant improvement. The analysis of **bias dependency on diet switches** appears to be **driven by artefacts of the experimental setup rather than biologically meaningful patterns** . It is imperative to clarify **which observed effects stem from methodological design choices and which reflect genuine ecological dynamics** . Similarly, the **case study on geese is incomplete** , as it lacks a **visual representation of the δ¹³C vs δ¹⁵N data** , making it difficult to evaluate the model outputs. This **visualisation must be included** , along with a **revised explanation of λT** , which is currently misdefined by conflating turnover rates with sampling frequency.

In addition to these substantive concerns, the manuscript requires **a major revision of its language and presentation** . The text contains **unclear phrasing, excessive technical jargon, and numerous grammatical inconsistencies** , all of which hinder readability. A **thorough linguistic revision** is necessary to meet publication standards. Furthermore, the **figures require extensive revision** —captions must be self-explanatory, and visual representations must be refined to more clearly illustrate key patterns. Where needed, **new figures should be generated** to better convey complex relationships, particularly in the bias analysis across different diet scenarios.

Given the extent of necessary revisions, **a resubmission will only be considered if all major concerns are fully addressed** . The revised manuscript **must include a detailed response letter** outlining precisely how each of the reviewers’ critiques has been incorporated. **Failure to adequately resolve these fundamental methodological and interpretative issues will result in the manuscript being deemed unsuitable for further consideration.**

I strongly encourage you to approach these revisions with the depth and rigour that this subject matter demands. If these fundamental concerns are **comprehensively addressed** , the study has the potential to make a **meaningful contribution to the field** . I look forward to receiving a **significantly improved** manuscript in due course.

**Sincerely,**

Przemysław Mroczek, Dr. hab.

Academic Editor

PLOS ONE

Reviewers' comments:

Reviewer's Responses to Questions

**Comments to the Author**

1. Is the manuscript technically sound, and do the data support the conclusions?

Reviewer #1: Yes

Reviewer #2: No

2. Has the statistical analysis been performed appropriately and rigorously? 

Reviewer #1: Yes

Reviewer #2: No

3. Have the authors made all data underlying the findings in their manuscript fully available?

Reviewer #1: Yes

Reviewer #2: Yes

4. Is the manuscript presented in an intelligible fashion and written in standard English?

Reviewer #1: Yes

Reviewer #2: No

5. Review Comments to the Author

Reviewer #1: This study sets out to quantify and analyze the bias associated with different stable isotope mixing model approaches, and compare the bias of standard static mixing models with models

incorporating information about turnover dynamics, and a dynamic model incorporating temporal isotopic dynamics that was developed within the study. I believe the questions adressed by the

paper are highly relevant and fill an important knowledge gap in isotope ecology by implicitly adressing the question of temporal dynamics, in particular variation in the lamda parameter.

The paper is overall well written, but could profit from a more streamlined discussion and a better incorporation of relevant literature.

The introduction is well written, and conveys the knowledge gaps and the relevance of the questions posed in the paper. However, it might be worthwhile to add a bit of information on the

difference between frequentist and Bayesian approaches (as the latter do allow to incorporate uncertainty in source isotopic composition to a degree, be it temporal or spatial), and the

development of the field as such.

The Methods and results are very detailed and contain all information necessary to follow the later discussion, maybe even too much information at times. Some paragraphs in the Methods

section read a bit like a mixture of introduction and Methods (I´ve indicated sections where I thought this to be the case below). Similarly, the Results sometimes interpret the data to

an extent that I would expect more for a discussion (also indicated below). The discussion of this paper is very long and could benefit from more focus at places. It also should

put the results of the study more into context with the existing literature. As it stands, the discussion largely discusses the validity of the models used, and repeats results that were

already shown in the results section. The paper of Phillips et al. (2014) is heavily cited in the discussion, but other literature on mixing models (e.g., Stock & Semmens; Canseco et al.)

is not discussed. Overall, I think the discussion could benefit from more structured, streamlined writing with a clearer focus on the relevant results and what they mean for the field.

All figures of this paper need a thorough revision! A figure caption should be able to stand on its own and state clearly what data was used for the figure and how it has to be interpreted.

I´ve included some more detailed comments below.

Specific comments:

L15: "quantify" instead of "decipher"?

L16: Maybe just write "These models assume", as they are STILL based on that assumption, not only historical (most mixing model studies still use SMMs, no?)

L21: relatively to what? Maybe delete?

L26: "isotopic integration"? Do you mean turnover?

L27: Maybe write "dietary changes" instead of "diet switch"? Could also be caused by changes in food availability than an active switching of sources.

L63: Stock & Semmens 2016 would also be an important paper to introduce here (unifying error structures to estimate variation)

L117-120: This section reads like an intro. Maybe introduce the model classes needed for your study in the intro and focus the Methods on a simpler structure of "to achieve X, we did Y"?

L131-134: See above, this is a lot of details on the "why" of this model, which a reader should already know after reading the intro.

L202-206: Is all this intro really needed to understand how you estimated bias?

L240-241: What do you mean by constrained here? No more than x tracer + 1 sources? Please clarify.

L322: "was then simulated"

L322-323: Most studies approach their study system with some prior knowledge of the seasonal dynamics. Were those 10 sampling dates chosen randomly, or did you set them at equal intervals?

L417-418: Sounds like a discussion point

L421: Why? Point of discussion, why is this the key observation?

L476-479: This is repeating your results, I would remove the figure references and just discuss the interpretation without repeating the results leading up to it.

L480-483: Not sure this is the right place to discuss the limitations. Rather put your findings into context with other studies.

L484-498: Possibly outside of the scope of this study, but I would be interested how a Bayesian MM would compare in the unconstrained cases?

L491-498: This section describing Phillips et al. recommendations on mixing polygons could maybe be summarized in one or two sentences to start off the next paragraph?

L505-515: Again, multiple repetitions of results in this section, which I don´t think are strictly necessary to understand the paper and inflate the length.

L516: I stopped commenting every repetition or reference to specific sections of the results after this one. My recommendation would be to remove most, if not all from your discussion.

L517-518: Outputs commonly observed, but you offer no examples. I would recommend to use way more literature to put your results into context, you already have a lot from your review, no?

L552: Maybe write out the parameter so readers that may have forgotten what lamda T means can still make something of the header

L616-627: This whole section feels reads like a repetition of your intro. I would recommend to either use the review for the intro (which works well I think) OR the discussion, not both.

L630-639: Why is this in conclusions, or rather, WHAT is the conclusion here?

L640-646: I think this section (barring the repetition of results at the end) would be a better leading paragraphs for the conclusions

L658-661: I think this lifetime perspective of mixing models would ONLY work if an all individuals can use the same resources along their lifetime. For gape-limited predators, a mixing

model over their lifetime would assume a juvenile (e.g., a newborn shark <50 cm) could, in theory, experience a contribution from the sources of an adult (i.e., very large prey items)

L667-672: Yes, but you didn´t assess variation in TDFs. This conclusion paragraph appears rather chaotic and unfocussed. I think it would be best to split it into a limitations paragraph,

in which you could discuss the limitations of YOUR models along with the things you didn´t look at, and a conclusions paragraph that specifically highlights the take home messages of

your study and outlines future directions.

Appendix 6: Maybe reiterate what were the simulated diets/isotopic systems.

Appendix 7: How do I see which space of the six isotopic spaces is which from the figure A, in order to reference it to figure B?

Reviewer #2: The review is uploaded as a separate attachment due to usage of special characters.

6. PLOS authors have the option to publish the peer review history of their article (what does this mean? ). If published, this will include your full peer review and any attached files.

**Do you want your identity to be public for this peer review?** For information about this choice, including consent withdrawal, please see our Privacy Policy .

Reviewer #1: No

Reviewer #2: **Yes: ** Maciej P. Lewicki

---

## [Author Response · Author response to Decision Letter 1]

27 May 2025

Every comments made by reviewers and the Editor were answered in the file "Response to reviewers" submited here.

---

## [Editor Report · Decision Letter 1]

18 Jun 2025

PONE-D-24-58834R1From Static to Dynamic: Embracing dynamics in Isotopic Diet EstimationPLOS ONE

Dear Dr. Cathelin,

Thank you for submitting your manuscript to PLOS ONE. After careful consideration, we feel that it has merit but does not fully meet PLOS ONE’s publication criteria as it currently stands. Therefore, we invite you to submit a revised version of the manuscript that addresses the points raised during the review process.

The authors have substantially improved the manuscript and addressed the majority of reviewer and editorial comments. However, they have not fully resolved Reviewer 2’s methodological concerns regarding uncertainty estimation and correlation structure, which are critical for transparency. To meet PLOS ONE’s standards, a final revision is required to clearly state these limitations and adjust any overgeneralised conclusions.

We look forward to receiving your revised manuscript.

Kind regards,

Przemysław Mroczek, Dr. hab.

Academic Editor

PLOS ONE

Journal Requirements:

**Additional Editor Comments:**

Dear Dr Cathelin and co-authors,

I have now carefully reviewed your revised manuscript entitled “From Static to Dynamic: Embracing dynamics in Isotopic Diet Estimation”, together with your detailed responses to both the reviewers’ and my editorial comments. I appreciate the substantial improvements you have made. Most of the key concerns from Reviewer 1 and my own editorial suggestions have been addressed thoroughly, and the manuscript is much clearer in its current form.

However, I note that several fundamental methodological criticisms raised by Reviewer 2 — particularly regarding the neglect of source correlations and the absence of uncertainty quantification — have not been fully addressed in the current version. While you have justified your choice of a deterministic framework and the lack of existing Bayesian dynamic models, these limitations must be presented very clearly to ensure transparency and to avoid overgeneralising the practical recommendations for real-world applications.

Therefore, I am inviting you to submit a revised version of the manuscript addressing the following specific points:

Explicitly acknowledge in the Discussion and Conclusions that your approach does not model uncertainty or correlation structures, unlike standard Bayesian mixing models. Please clarify that your results are based on central estimates only and that this may limit the interpretation of bias magnitude.

Adjust any wording in the manuscript that could imply that the dynamic mixing model is fully robust for real-world application without addressing statistical uncertainty. Recommendations should be nuanced to reflect this limitation.

Consider adding a short statement in the Data Availability and Methods sections noting that the provided scripts support deterministic replication but do not implement uncertainty propagation or MCMC sampling.

These clarifications are important to align the manuscript with PLOS ONE’s requirements for methodological transparency and to properly inform readers about the scope and limitations of the work.

Please submit your revised manuscript and provide a concise response letter summarising how these final points have been addressed.

Thank you once again for your valuable contribution and your constructive responses throughout the review process.

Kind regards,

Przemysław Mroczek, Dr hab.

Academic Editor

---

## [Author Response · Author response to Decision Letter 2]

15 Jul 2025

All remaining editor comments are adressed in the "Response to Reviewers" document in the submitted manuscript.

---

## [Editor Report · Decision Letter 2]

17 Jul 2025

PONE-D-24-58834R2From Static to Dynamic: Embracing dynamics in Isotopic Diet EstimationPLOS ONE

Dear Dr. Cathelin,

Thank you for submitting your manuscript to PLOS ONE. After careful consideration, we feel that it has merit but does not fully meet PLOS ONE’s publication criteria as it currently stands. Therefore, we invite you to submit a revised version of the manuscript that addresses the points raised during the review process.

The manuscript meets all scientific and editorial requirements and the authors have adequately addressed all prior comments. However, in line with PLOS ONE’s data policy and best practices for reproducibility, a stable, citable version of the code must be archived in a DOI-minting repository such as Zenodo prior to acceptance, as GitHub alone does not guarantee long-term accessibility or version control.

We look forward to receiving your revised manuscript.

Kind regards,

Przemysław Mroczek, Dr. hab.

Academic Editor

PLOS ONE

Journal Requirements:

Additional Editor Comments :

Dear Dr Cathelin,

Thank you for your thorough and constructive revision of the manuscript entitled "From Static to Dynamic: Embracing Dynamics in Isotopic Diet Estimation".

I have carefully reviewed the revised version and I am pleased to confirm that the manuscript now meets the scientific and editorial standards required for publication in PLOS ONE. The changes you have implemented are appropriate and well executed, and your responses to the previous editorial comments are satisfactory.

However, in accordance with PLOS ONE’s data availability policy, which requires that all data and code necessary to replicate the findings be deposited in a stable, long-term public repository, I must request a final revision before acceptance. Specifically, although the code is currently available on GitHub, this platform does not guarantee permanence or version control with a persistent identifier. To ensure long-term accessibility and the reproducibility of the results reviewed by the referees, I kindly ask that you deposit the current version of the code in a DOI-minting repository such as Zenodo or an equivalent archive prior to acceptance. Please also update the Data Availability Statement in the manuscript accordingly.

I look forward to receiving the updated version and to proceeding with acceptance shortly thereafter.

Kind regards,

P. Mroczek

---

## [Author Response · Author response to Decision Letter 3]

21 Jul 2025

We thank you for the last review of our work. As requested we created a repository in Zenodo with all our codes and data to ensure reproductibility. The link was changed in the manuscript and the Data Availability Statement.

---

## [Editor Report · Decision Letter 3]

1 Aug 2025

From Static to Dynamic: Embracing dynamics in Isotopic Diet Estimation

PONE-D-24-58834R3

Dear Dr. Cathelin,

We’re pleased to inform you that your manuscript has been judged scientifically suitable for publication and will be formally accepted for publication once it meets all outstanding technical requirements.

Kind regards,

Przemysław Mroczek, Dr. hab.

Academic Editor

PLOS ONE

Additional Editor Comments (optional):

Dear Authors,

I have carefully reviewed the new version of your manuscript entitled "From Static to Dynamic: Embracing dynamics in Isotopic Diet Estimation". I acknowledge the changes introduced, including the addition of the full dataset and code to a public Zenodo repository, as well as improvements in the clarity and presentation of the dynamic mixing model framework. These modifications appropriately address the previous editorial comments.

Although the revised version contains only minor textual changes, the manuscript is now clear, well-organised, and scientifically sound. I appreciate your thoughtful response and the high quality of your work throughout the review process.

Thank you again for your cooperation.

With best regards,

P. Mroczek
---

## [Editor Report · Acceptance letter]

PONE-D-24-58834R3

PLOS ONE

Dear Dr. Cathelin,

I'm pleased to inform you that your manuscript has been deemed suitable for publication in PLOS ONE. Congratulations! Your manuscript is now being handed over to our production team.

Kind regards,

on behalf of

Dr. hab. Przemysław Mroczek

Academic Editor

PLOS ONE